# A Novel Route to High-Quality Graphene Quantum Dots by Hydrogen-Assisted Pyrolysis of Silicon Carbide

**DOI:** 10.3390/nano10020277

**Published:** 2020-02-06

**Authors:** Na Eun Lee, Sang Yoon Lee, Hyung San Lim, Sung Ho Yoo, Sung Oh Cho

**Affiliations:** Department of Nuclear and Quantum Engineering, Korea Advanced Institute of Science and Technology (KAIST), Daejeon 34141, Korea; pancy6@kaist.ac.kr (N.E.L.); sangyoonlee@kaist.ac.kr (S.Y.L.); samsterdam@kaist.ac.kr (H.S.L.); nafirst4@kaist.ac.kr (S.H.Y.)

**Keywords:** graphene quantum dots, silicon carbide, hydrogen-assisted pyrolysis, high-quality

## Abstract

Graphene quantum dots (GQDs) can be highly beneficial in various fields due to their unique properties, such as having an effective charge transfer and quantum confinement. However, defects on GQDs hinder these properties, and only a few studies have reported fabricating high-quality GQDs with high crystallinity and few impurities. In this study, we present a novel yet simple approach to synthesizing high-quality GQDs that involves annealing silicon carbide (SiC) under low vacuum while introducing hydrogen (H) etching gas; no harmful chemicals are required in the process. The fabricated GQDs are composed of a few graphene layers and possess high crystallinity, few defects and high purity, while being free from oxygen functional groups. The edges of the GQDs are hydrogen-terminated. High-quality GQDs form on the etched SiC when the etching rates of Si and C atoms are monitored. The size of the fabricated GQDs and the surface morphology of SiC can be altered by changing the operating conditions. Collectively, a novel route to high-quality GQDs will be highly applicable in fields involving sensors and detectors.

## 1. Introduction

Graphene quantum dots (GQDs), composed of few graphene layers with a size less than 30 nm, have attracted research interest due to their unique properties, such as possessing a large surface area, low toxicity and strong and tunable photoluminescence [1,2,3,4]. With these advantageous characteristics, GQDs can be used for bioimaging [5,6], biosensing [7], photovoltaics [8,9], drug delivery [10,11] and optoelectronic devices [12,13]. Both top-down and bottom-up approaches have been developed to synthesize GQDs. The top-down approaches include electron-beam lithography [4], chemical oxidation [14,15], electrochemical exfoliation [7,8,16], hydrothermal/solvothermal treatment [6,17] and microwave/ultrasound methods [18,19,20,21], all of which have difficulties in precisely controlling the size of the GQDs produced and in reducing the defects of synthesized GQDs [22]. Bottom-up approaches rely on the carbonization of organic precursors [23,24,25,26,27], but these approaches face issues resulting from the involvement of harmful chemicals and the complexity of the synthesis process [22]. Additionally, the basal plane or edges of GQDs synthesized by these methods are passivated by oxygen functional groups such as carboxyl, carbonyl and hydroxyl groups. The characteristics of GQDs, such as having an effective charge transfer and quantum confinement, are hindered by these defects [4,28]. Currently, only a few studies have reported fabricating high-quality GQDs; thus producing GQDs that have high crystallinity, few defects and high purity is therefore of great importance as GQDs have many promising applications in fields involving sensors and detectors [4,28].

In this study, we present a novel route to fabricating high-quality GQDs that involves hydrogen-assisted pyrolysis of silicon carbide (SiC) without the use of harmful chemicals. The pyrolysis of SiC has been used widely for the synthesis of epitaxial graphene (EG) [29,30]. Generally, EG is synthesized in two completely different environments: either under ultrahigh vacuum (UHV, <10^−10^ Torr) at comparatively low temperature (<1300 °C) or under atmospheric pressure in argon (Ar) gas at relatively high temperature (>1600 °C) [31]. In the UHV environment, irregular pits are formed on the SiC surface and small graphene clusters (of a few hundreds of nanometers in size) are created on the pitted surface. In comparison, in the Ar atmosphere environment, a uniformly stepped SiC surface is formed and large EG clusters (> hundreds of μm in size) are synthesized on the stepped surface. Before thermal decomposition, the SiC surface is treated by hydrogen (H) etching. The H etching process also plays an important role in determining the morphology of the fabricated EG [32]. In our previous research, we discovered that if a SiC plate is H etched under vacuum (~100 Torr), steps of heights reaching up to 100 nm could be fabricated [33]. These findings indicate that the morphologies of both the graphene and SiC surface are significantly affected by the annealing conditions, and particularly the etching rate of Si and C atoms on the SiC surface. These results inspired us to develop and propose a novel approach to synthesizing GQDs by annealing SiC under low vacuum while introducing H etching gas into the vacuum environment. When pyrolyzed under vacuum containing the etching gas, very rapid etching of SiC occurs, converting the surface into a bumpy structure with pits that are a few micrometers deep. On this bumpy surface, an abundance of H-terminated high-quality GQDs is ubiquitously created. Fabricated GQDs have a highly ordered crystalline structure with high purities. The size of the synthesized GQDs can be controlled by adjusting the operating conditions, such as the annealing temperature.

## 2. Materials and Methods

### 2.1. Materials

We purchased 4H nitrogen-doped SiC plates with an off-axis angle of 4° relative to the (0001) basal plane from TankeBlue Co., Ltd. (Beijing, China). Ethanol (C_2_H_5_OH, > 99.9%) was purchased from Merck Chemicals (Darmstadt, Germany).

### 2.2. High-Quality GQDs Preparation

The 4H nitrogen-doped SiC plates were ultrasonically cleaned in ethanol and subsequently annealed in an alumina furnace, where a mixed gas comprising argon (Ar, 96 at.%) and hydrogen (H, 4 at.%) flowed through. The pressure of the vacuum furnace was changed from 80~160 mTorr by adjusting the flow rate of the mixture gas from 8 to 16 sccm. The maximum temperature of the furnace was also modified from 1400~1500 °C. The heating rate of the furnace to reach the maximum temperature was 5 °C per min, the maximum temperature was maintained for 30 min and the cooling rate to reach room temperature was 5 °C per min. The annealed SiC samples were sonicated at 80 kHz in 2 mL of ethanol for 3 min to detach the synthesized GQDs from the SiC substrates. Subsequently, the solutions of the GQDs were centrifuged at 8000 rpm with 10,000 molecular weight cut-off (MWCO) microfilters to eliminate any large particles that were detached from the substrates.

### 2.3. Characterizations

The surface morphology of the specimens was characterized using a field-emission scanning electron microscope (FESEM, Hitachi S-4800, Hitachi, Tokyo, Japan). The size distribution and lattice spacing of the GQDs were examined using a transmission electron microscope (TEM) and high-resolution transmission electron microscope (HRTEM; Tecnai G² F30 S-Twin, FEI, Hillsboro, OR, USA). The thickness of the GQDs was determined using an atomic force microscope (AFM; XE70, Park systems, Suwon, Korea). The Raman spectra were measured using a Raman spectrophotometer (Resolution of 0.75 cm^−1^, Horiba Jobin Yvon, Kyoto, Japan) with a 514 nm laser source with a spot size of 2 μm. The chemical composition of the GQDs was analyzed by a Fourier transform infrared (FT-IR) spectrometer (Nicolet iS50, Thermo Fisher Scientific Instrument, Waltham, MA, USA). The X-ray photoelectron spectroscopy (XPS) spectra of the GQDs were measured on a gold substrate using a 3000 W Al Kα as the microfocused monochromatic X-ray source (K-alpha, Thermo VG Scientific, Waltham, MA, USA).

## 3. Results and Discussion

### 3.1. Morphological Features

GQDs were prepared via the thermal decomposition of the SiC plates, as shown in Figure 1. The plates initially had flat surfaces, as shown in Figure 2a. However, when the plates were annealed at 1500 °C for 30 min in a vacuum furnace containing a gas mixture comprising Ar (96 at.%) and H (4 at.%), the morphology of the SiC surface was completely altered, dependent on the vacuum pressure, which was controlled by adjusting the flow rate of the gas mixture. At the vacuum pressure of 160 mTorr, irregularly shaped particulates of a few hundred nanometers in size were formed on the flat SiC surface, as can be seen in the FESEM image as shown in Appendix A. Decreasing the pressure to 80 mTorr led to the flat surface being converted into a bumpy structure with a surface roughness of a few micrometers, and on this bumpy surface, a large number of nanoparticles were created, as shown in Figure 2b and Appendix A. The nanoparticles were easily detached from the SiC plate by sonication in ethanol. The amount of GQDs detached from the SiC plate of 1cm × 1cm was about 200 μg. The TEM image shows that the nanoparticles were monodispersed with a size of 2.58 ± 0.31 nm, as shown in Figure 2c and Appendix A).

### 3.2. Structural Features

The HRTEM image shows that the nanoparticles have a highly-ordered crystalline structure with a lattice spacing of 0.21 nm, as shown in the inset of Figure 2c, which corresponds to that of the (100) planes of graphite [34]. In addition, the Raman spectra of the nanoparticles exhibit the D band at 1348 cm^−1^, the G band at 1582 cm^−1^ and the 2D band at 2701cm^−1^, as shown in Figure 3a. Both the TEM and Raman measurements support that the nanoparticles are indeed GQDs. The D to G peak intensity ratio (I_D_/I_G_) of the GQDs is 0.79, and the 2D to G peak intensity ratio (I_2D_/I_G_) is 0.64. Both values indicate a relatively high crystallinity considering that a large number of disordered edges exist on the GQDs [28]. It is notable that such a high 2D band, as seen in this study, has rarely been reported for previously synthesized GQDs [16,35,36]. The AFM image shows that the average thickness of the GQDs is ~0.972 nm, as shown in Figure 3b, indicating that the GQDs synthesized have few layers. The XPS survey spectrum, as shown in Appendix A, exhibits an intense C peak in the absence of any other peaks, such as that of oxygen, which indicates that the fabricated GQDs are high in purity. The high-resolution C1s XPS spectrum also supports that the GQDs are indeed highly pure. The spectrum exhibits only two peaks, corresponding to those of aromatic sp^2^ (284.4 eV) and sp^3^ (285 eV) carbon, with no peaks that are attributable to other elements, such as oxygen and nitrogen observed, as shown in Figure 3c, reflecting the high purity state of the fabricated GQDs. The FT-IR spectrum shows several peaks stemming from the aromatic C=C (1550, 1650 cm^−1^) and C–H (2854, 2923 cm^−1^) bonds, as shown in Figure 3d. The highly prevalent 2D band, the clear lattice spacing and the lack of impurities indicate that the synthesized GQDs are of high quality, with only a few defects on the basal plane. Additionally, the presence of sp^3^ carbon and C–H bonds, supported by the obtained XPS and FT-IR spectra, suggests that the edges of GQDs are terminated with hydrogen. Consequently, from these analyses, we can conclude that high-quality GQDs with hydrogen-terminated edges are produced by annealing the SiC under low-vacuum environment containing H gas. The size of the GQDs can be controlled by adjusting the annealing temperature; the average size of the GQDs gradually increased from 2.58 to 5.20 nm when the temperature was decreased from 1500 °C to 1400 °C at a constant pressure of 80 mTorr, as shown in Figure 4.

### 3.3. The Formation Mechanism of High-Quality GQDs

The formation mechanism of the GQDs can be elaborated as follows and shown in Figure 1. The process of synthesizing the GQDs is very similar to that of EG on SiC. EG is generally synthesized by H etching of the SiC surface, followed by the thermal decomposition of the etched SiC [31]. H etching is carried out under atmospheric pressure in an Ar and H gas mixture, resulting in an array of steps to form on the SiC surface with heights less than 1.5 nm [37]. The step structure assists in the generation and growth of large and uniform graphene. However, in our study, H etching was not performed prior to the GQDs synthesis. Instead, the H etching and thermal decomposition were carried out simultaneously by annealing the SiC plates under low vacuum (mTorr range) while introducing H gas into the vacuum environment. The annealing environment significantly affects the morphologies of both the SiC surface and the synthesized graphene [31,32]. During annealing, H removes Si and C atoms from the SiC surface by forming volatile silicon hydrides and hydrocarbons. However, the removal rate of C is higher than that of Si due to hydrocarbon having a lower formation free energy than silicon hydride [38]. Additionally, annealing also induces sublimation of Si and C from the SiC surface with Si subliming faster than C due to the former having a lower vapor pressure than the latter [39]. If SiC is annealed in a vacuum environment containing H, both Si and C atoms are more rapidly removed from the SiC surface, leading to faster etching when compared to annealing in an atmospheric pressure environment containing H, or in a UHV environment without H. This phenomenon is attributed to the fact that the decrease in vacuum pressure increases the Si sublimation rate, leading to a more rapid exposure of the C layer; the exposed C layer is then etched away by the H. However, if the vacuum pressure is too low, only a very small amount of H gas can be present, resulting in the C atoms not being readily removed; in contrast, the Si atoms are being rapidly sublimated away. Consequently, if a SiC plate is annealed in the presence of H gas at a pressure that falls within a specific range, vigorous etching of the SiC occurs, leading to the formation of a very rough surface, like the one with the bumpy structure shown in Figure 2b.

Simultaneously with the etching process, graphene is synthesized on the bumpy SiC surface. C atoms are left on the SiC surface leading to graphene nucleation and growth due to the aforementioned annealing-induced Si sublimation [40]. However, the vigorous H etching process suppresses the growth of graphene, because C atoms are removed by the reactive H [41]. As a consequence, only small-sized GQDs are fabricated. H also reacts with the dangling bonds at the edges of the GQDs, resulting in H-terminated GQDs [42]. Meanwhile, increasing the annealing temperature enhances the etching rate, resulting in even smaller GQDs, as shown in Figure 4 [43]. In contrast, increasing the vacuum pressure decreases the etching rate, leading to the formation of larger graphene crystallites and a less rough SiC surface. When the vacuum pressure is increased from 80 mTorr to 120 mTorr, the size of graphene crystallites increases from 2.58 nm to hundreds of nm and the surface roughness of SiC decreases from a few micrometers to less than 1 μm, as shown in Appendix A. When the pressure is further increased to 160 mTorr, graphene with a size in the order of micrometers is produced on a flat pitless SiC surface, as shown in Appendix A. The concentration of H in the H/Ar gas mixture can also affect the size of the resulting GQDs; however, the adjustment of the concentration, unless drastic, does not lead to a significant size variation, thus the effect of hydrogen concentration on the size of the GQDs was ignored.

## 4. Conclusions

In summary, we have presented a novel method for the fabrication of high-quality GQDs that involves the hydrogen-assisted pyrolysis of SiC in a vacuum. Annealing in a vacuum environment containing hydrogen causes a vigorous etching of Si and C atoms on the SiC surface, resulting in the formation of a bumpy surface and the production of hydrogen-terminated high-quality GQDs. The synthesized GQDs have a highly-ordered crystalline structure with almost no defects or impurities. The size of the GQDs can be altered by changing the operating conditions, such as the annealing temperature and the vacuum pressure. Thus, we believe that our proposed approach will be highly applicable in fields involving detectors and sensors.

## Figures and Tables

**Figure 1 nanomaterials-10-00277-f001:**
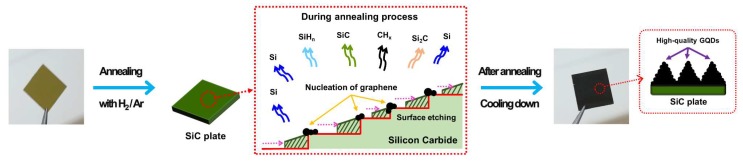
The schematic layout of the synthesis of high-quality graphene quantum dots (GQDs) by the hydrogen-assisted pyrolysis of silicon carbide (SiC).

**Figure 2 nanomaterials-10-00277-f002:**
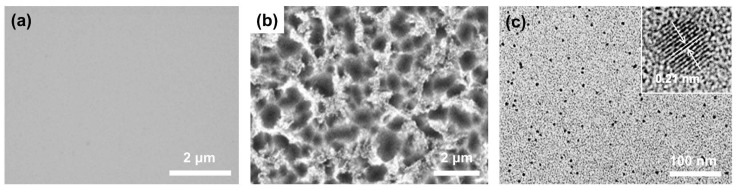
The field-emission scanning electron microscope (FESEM) image of (**a**) the pristine SiC plate and (**b**) the GQDs on the SiC plate after being annealed at 1500 °C on hydrogen etching gas. (**c**) The transmission electron microscope (TEM) image of the detached GQDs and the high-resolution transmission electron microscope (HRTEM) image of the GQDs with their lattice spacing.

**Figure 3 nanomaterials-10-00277-f003:**
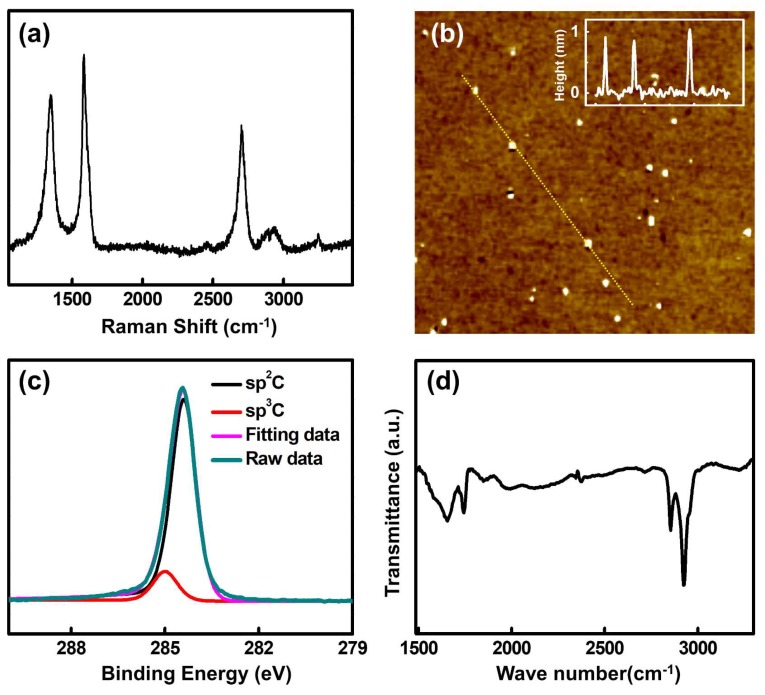
(**a**) The Raman spectra of the detached GQDs. (**b**) The atomic force microscope (AFM) image of the GQDs with the thickness graph of GQDs in yellow (the length of the yellow line inside the AFM image is 2.7 μm). (**c**) The X-ray photoelectron spectroscopy (XPS) high-resolution C1s spectrum and (**d**) the Fourier transform infrared (FT-IR) results of the GQDs.

**Figure 4 nanomaterials-10-00277-f004:**
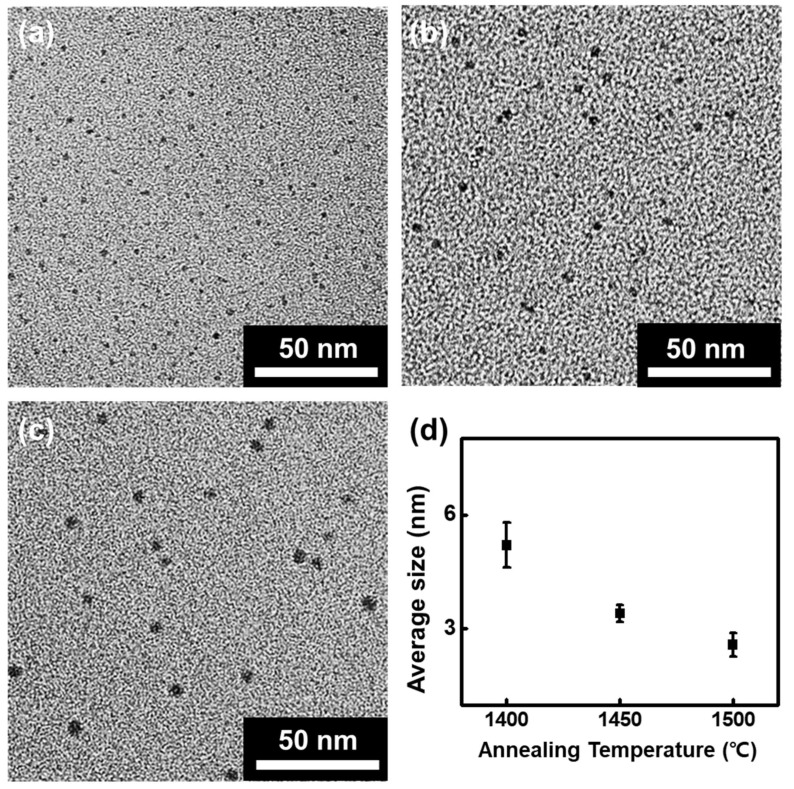
The HRTEM image of the GQDs fabricated on the SiC at a temperature of: (**a**) 1500 °C, (**b**) 1450 °C, (**c**) 1400 °C, (**d**) The correlation graph between the annealing temperature and the average size of the GQDs.

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
