# Peer review of "A Novel Route to High-Quality Graphene Quantum Dots by Hydrogen-Assisted Pyrolysis of Silicon Carbide"

_nanomaterials, 2020, doi:10.3390/nano10020277_

Round 1

Reviewer 1 Report

The authors report the preparation of graphene QDs (GQDs) via hydrogen-assisted pyrolysis of SiC. The formation mechanism of GQDs is also discussed. From my opinion, the synthesis method requires drastic conditions and cannot easily be implemented in labs to equiped. The authors must demonstrate that they produced high quality GQDs, for example by providing the optical properties of the nanocrystals.

Some of the results presented are of interest. The authors should consider the following comments before submitting a revised version.

the introduction should be revised. The authors should indicate that the presence of CO2H, OH and C=O groups at the surface or the periphery of GQDs allows to increase the water dispersibility of the dots. Moreover, these functional groups allow to tune the PL emission wavelength (from blue-green to orange) some recent examples describing the use of GQDs should be added to demonstrate the potential of these nanomaterials (Materials Science and Engineering C 2019, 103, 109824; Biomaterials 2019, 206, 61-72; Nanomaterials 2020, 10, 104; ...). paragraph 2.2 : indicate the amount of GQDs (or the yield) that can be produced in a typical synthesis. can the authors comment on the dispersity of their GQDs in solvents ? it would be of interest to add UV-visible absorption, the PL emission spectrum and the PL QY of GQDs figure 2c : add a size distribution

Author Response

We upload an attachment to response to reviewer's comment. Please see the attachment.

Reviewer 2 Report

In the current manuscript the authors show results of graphene nanoparticles using hydrogen-assisted pyrolysis of silicon carbide. The authors rigorously characterize the nanoparticles obtained as a result of synthesis using several independent techniques. The manuscript is well written; the presentation of the results is clear and concise. I recommend it for publication after addressing the points listed below.

 In my view, one of the key points of the work that could be discussed in more details is narrow size distributions of the particles. It is surprising for me that a relatively small change of temperature results in such a clear change of the average size, while keeping the width of the distribution so narrow. I believe that researchers working in this field would be interested to see the dependence between the size distribution (maximum and width) and annealing temperature (or pressure) in a broader range. Are there other factors that may have a similar effect on the particle size in the current method?

Reviewer 3 Report

This submission deals with the innovative synthesis of multi-layers Graphene quantum dots (GQDs), featured with high purity and crystallinity, which are yielded by SiC annealing, under low vacuum, H2-containing, gas pressure. Products have been exhaustively characterised by different microscopy techniques (SEM, TEM, AFM) vibrational (Raman, FT-IR) and XPS spectroscopy. The topic is well introduced by the Authors and the conclusions of interest are well supported by experimental data. This work is appropriate for the Readership of Nanomaterials and can be published after some minor revisions.

Although this MS is well written, in page 2, lines 83-85, the experimental yield of GQDs should be rephrased, because it is not too much understandable;  In Figure 1, SiHn silanes etching should be also added, besides sublimation; Albeit not mandatory in this measurements, however the actual experimental Raman resolution for the reported measurements should be stated; In some of the microscopy techniques (e.g., SEM-EDAX, Mater Sci Engineer B 2006 131 72) C elemental analysis might support other Raman characterisation. 

Round 2

Reviewer 1 Report

All corrections were done by the authors. The manuscript can be accepted by Nanomaterials.